# Prostate Apoptosis Response-4 (Par-4): A Novel Target in Pyronaridine-Induced Apoptosis in Glioblastoma (GBM) Cells

**DOI:** 10.3390/cancers14133198

**Published:** 2022-06-29

**Authors:** Jeevan Ghosalkar, Vinay Sonawane, Tejal Pisal, Swati Achrekar, Radha Pujari, Ashish Chugh, Padma Shastry, Kalpana Joshi

**Affiliations:** 1Cell Biology Division, Cipla Ltd., Vikhroli, Mumbai 400083, India; dr.jeevan@cipla.com (J.G.); vinay.sonawane@cipla.com (V.S.); tejal.pisal@cipla.com (T.P.); swati.achrekar@cipla.com (S.A.); 2National Centre for Cell Sciences (NCCS), Pune 411007, India; radhapujari@gmail.com; 3Department of Neurosurgery, D.Y. Patil Medical College, Pune 411018, India; apchugh@yahoo.com

**Keywords:** glioblastoma (GBM), pyronaridine, repurposing, apoptosis, prostate apoptosis response-4 (Par-4), therapeutic target

## Abstract

**Simple Summary:**

GBM treatment is an area of high unmet need due to the heterogeneous and anaplastic character of this cancer in turn leading to an extremely poor prognosis. Finding new molecular entities by traditional or de novo approaches to drug discovery is lengthy and expensive. Repurposing existing drugs can be attractive as the process is often less risky, more cost, and time-effective. Amongst potential drug-repurposing candidates, Pyronaridine (PYR), an antimalarial drug has shown anti-cancer effects against several cancers, however, its potential for the treatment of GBM has not been explored. In this study, we have identified a unique mechanism of action of PYR against GBM by upregulating a tumor suppressor protein, Par-4 along with the elucidation of the complex network of pathways mediated through Par-4 leading to GBM cell death.

**Abstract:**

Glioblastoma (GBM) is an aggressive form of brain tumor with a median survival of approximately 12 months. With no new drugs in the last few decades and limited success in clinics for known therapies, drug repurposing is an attractive choice for its treatment. Here, we examined the efficacy of pyronaridine (PYR), an anti-malarial drug in GBM cells. PYR induced anti-proliferative activity in GBM cells with IC_50_ ranging from 1.16 to 6.82 µM. Synergistic activity was observed when PYR was combined with Doxorubicin and Ritonavir. Mechanistically, PYR triggered mitochondrial membrane depolarization and enhanced the ROS levels causing caspase-3 mediated apoptosis. PYR significantly decreased markers associated with proliferation, EMT, hypoxia, and stemness and upregulated the expression of E-cadherin. Interestingly, PYR induced the expression of intracellular as well as secretory Par-4, a tumor suppressor in GBM cells, which was confirmed using siRNA. Notably, Par-4 levels in plasma samples of GBM patients were significantly lower than normal healthy volunteers. Thus, our study demonstrates for the first time that PYR can be repurposed against GBM with a novel mechanism of action involving Par-4. Herewith, we discuss the role of upregulated Par-4 in a highly interconnected signaling network thereby advocating its importance as a therapeutic target.

## 1. Introduction

Glioblastoma multiforme (GBM) is a highly heterogeneous and aggressive disease of all types of gliomas [1,2,3]. GBM remains an incurable disease with an incidence of nearly 6 cases per 100,000 people worldwide and more than 14,000 new cases are diagnosed in the US per year [4]. Even with the improvements in treatment modalities, survival and mortality figures remain dismal, only 25% of patients survive more than a year and only 5% of patients survive more than 5 years [5,6]. Surgery, chemotherapy, and radiotherapy remain the pillars to treat GBM. However, the intracranial site, aggressive biological behavior, and infiltrative growth of the disease present potential problems for the disease treatment [7]. Furthermore, the tumor recurrence is universal, and patients with recurrent tumors survive less than a year. Multiple factors such as glioma stem cells (GSC), genetic mutations, epigenetic modifications, tumor microenvironment (TME), and dysregulated pathways are collectively responsible for treatment resistance and tumor recurrence [8]. Considering the limited success of known therapies in the clinic, high cost (~1.8 billion USD), and long time frame (13.9 Y) for new drug development, repurposing existing drugs could be a better option for rapid clinical impact at a lower cost and time than de novo drug development [9].

In this study, we report the repositioning of Pyronaridine (PYR) against GBM. Pyronaridine, a benzonaphthyridine derivative is used presently in combination with artesunate (180/60 mg, PO, qdx3) against uncomplicated malaria and is safe [10,11]. It acts by inhibiting hemoglobin metabolism and prevents the formation of beta-hematin, leading to the buildup of toxic heme within the parasite, which succumbs to its death [12,13]. PYR is also known for its in vitro activity against SARS-CoV-2, Influenza, Ebola, Resistant TB, and Zoonotic diseases by a different mechanism of action [14]. A growing amount of evidence suggests that PYR could be pharmacologically targeted against cancer [15,16,17,18]. However, its molecular mechanism of action is not explored in-depth and thus remains poorly understood. One line of evidence suggests that PYR is structurally similar to compounds such as quinacrine and amsacrine, which are known to interfere in the metabolism of nucleic acids through intercalation of DNA and inhibition of topoisomerase II (Topo-II) [19,20], likewise, PYR is also stated to intercalate DNA as evident from viscometry studies performed with supercoiled Plasmid DNA. Thus, PYR could cause inhibition of cell proliferation and induction of apoptosis in multiple cancer cells [14,15]. Additionally, studies have shown that PYR being a P-gp inhibitor reverses the multi-drug-resistance (MDR) phenotype in MDR cancer cell lines, enhancing the accumulation and cytotoxic effect of anthracycline and Doxorubicin [21,22].

Not only is this the first study to report the efficacy of PYR in difficult-to-treat GBM, but importantly, this is also the first account highlighting a novel mechanism of action of PYR as a potent inducer of Par-4, a tumor suppressor protein. Par-4 is often down-regulated in several cancers including GBM, conferring resistance to treatment therapy [23,24,25]. Par-4 is known to cause apoptosis selectively in cancer cells but not in normal cells. Depending on the nature of the stimulus, Par-4 mediated apoptosis can occur via two different pathways, intrinsic and extrinsic [26]. The extracellular secretion of Par-4 is exhibited by both cancers as well as normal cells. The paracrine activity of Par-4 is mainly mediated through its interaction with GRP-78, which is often upregulated in cancer cells [27]. While in cancer cells, the extrinsic pathway can get triggered due to ER stress, additionally, small molecule secretagogues can be used to induce secretion of Par-4 by both normal as well as cancer cells [28,29,30]. This present study has led to the identification of PYR as a potent secretagogue of Par-4. Markedly, our data demonstrate the down-regulation of Par-4 in GBM patients as compared to the healthy volunteers further confirming the role of PYR as a potential candidate against GBM.

In addition to this notable finding, the mechanism involving PYR-mediated Par-4 induction has confirmed significant modulation of proliferation, hypoxia, EMT, apoptosis, and stemness markers. Taken together, compelling evidence generated during this study suggests that PYR could be a mechanistically suitable candidate to treat resistance/recurrence in GBM. We believe that the combination of inhibitors of multiple pathways and modulation of tumor suppressors could provide insight into potentially active drug combinations for future treatment of GBM.

## 2. Materials and Methods

### 2.1. Cell Lines and Reagents

The human glioma cells line viz, LN-18, U87MG, SNB-19, and U-251 were procured from ATCC and cultured in ATCC-recommended complete media. Whereas human neural glial cell-line (HNGC-2) and primary culture derived from GBM tumors (G1) were generated at National Centre for Cell Science (NCCS), Pune. Experimental methods on established patient-derived xenograft (PDX) cells CNXF-2599L were conducted at Charles River, Germany. The cells were maintained in a humidified chamber at 37 °C and 5% CO_2_. All the compounds were dissolved in dimethyl sulfoxide (DMSO) at a concentration of 10 mM and diluted in a culture medium immediately before use. Pyronaridine tetraphosphate (#VIA-404019) was procured from Chemical Centre (Mumbai, India) while Doxorubicin and Ritonavir were available in-house. Dulbecco’s phosphate-buffered saline without calcium and magnesium (#14025092), cell culture media, and FBS (#26140079) were purchased from Gibco (Thermofisher, Waltham, MA, USA). Lookout mycoplasma PCR kit (#MP0035) and PMS (#P9625) were from Sigma while 96-well flat-bottomed white polystyrene plates were from Corning (#3912) and MTS reagent from Promega (Madison, WI, USA, #G111).

### 2.2. In Vitro Cytotoxicity

Human glioma, PDX, HNGC-2, and G1 cells were seeded in 96 well-plates (2000–3000 cells/well) and kept at 37 °C at 5% CO_2_. Post 24 h incubation, PYR alone or in combination with other test compounds were added in triplicates in the wells. After 72 h of incubation, 20 µL of MTS: PMS reagent or CellTiter-Blue reagent was added to the wells. Cells were incubated for 4 h and the formation of colored tetrazolium product was measured at 490 nm or relative fluorescence (RFU) was measured at Ex/Em (λ = 570/600 nm) using a Spectramax ID5 plate reader (Molecular devices, San Jose, CA, USA). The results were calculated as % T/C over control. Half-inhibitory concentration (IC_50_) values were computed with the help of GraphPad Prism using a 4-parameter sigmoidal curve. The bliss index (BI) was calculated to determine the synergy or additive effect of drug combinations.

### 2.3. Spheroid Assay

LN-18 and U87MG cells were seeded into a U-shaped 96-well plate (5000 cells/well) in appropriate media. Spheroids were generated using a liquid overlay cultivation technique. Before seeding, 96-well plates were layered with 1% agarose under sterile conditions and kept undisturbed for an hour. Aggregation of cells in 96-well plates was facilitated by slowly rotating 96-well plates on the plain surface. Upon cell seeding single-cell suspension, the plate was incubated at 37 °C for 24 h. The spheroids were formed as a cluster of cells in the center of each well. The spheroids formed were treated with different concentrations of PYR for 5–6 days. After treatment, spheroids were stained with Alamar blue, and fluorescence was read at 530/590 nm. The results were calculated as % T/C over control and IC_50_ values were computed.

### 2.4. Cell Cycle Analysis and FITC-Annexin Assay

The effect of PYR on glioma cells was analyzed for ploidy content using flow cytometry. U87MG and LN-18 cells (~1 × 10^6^ cells per well) were seeded in six-well plates and further incubated at 37 °C and 5% CO_2_ for 24 h. Cells were treated with PYR at indicated concentrations. After treatment, cells were trypsinized, centrifuged at 400× *g* for 10 min, and washed twice with cold PBS. Further, cells were fixed with chilled 70% ethanol at 25 °C for 30 min. Ethanol fixed cells were centrifuged at 1000 rpm for 5 min and washed twice with PBS. Finally, cells were resuspended in PBS containing RNase A (20 µg/mL) and propidium iodide (PI, 50 µg/mL) and incubated at 25 °C for 45 min. Ten thousand cells were acquired using a flow cytometer (BD FACSLyric, BD Biosciences, San Jose, CA, USA) followed by data analysis on BD FACSuite software (Version 2.1, BD Biosciences, San Jose, CA, USA). The analysis area was gated for the selection of the cell population. The peak channels were marked and the % population was obtained for each phase of the cell cycle.

FITC-Annexin V apoptosis assay (#556547, BD Biosciences, San Jose, CA, USA) was performed as per the manufacturer’s instruction to determine early apoptosis. U87MG and LN-18 cells (~1 × 10^6^) were seeded in six-well culture plates. After treatment with PYR, U87MG and LN-18 cells were trypsinized and resuspended in PBS. Approximately 1 × 10^6^ cells were harvested and washed twice with cold PBS, then suspended in 500 µL binding buffer. A total of 10 µL Annexin V-FITC and 10 µL PI were added to the solution and mixed well. After 15 min of incubation, 10,000 cells were acquired and further analyzed using flow cytometry analysis.

### 2.5. Mitochondrial Potential (ΔΨM)

Mitochondrial transmembrane potential (ΔΨM) was studied using Mitoprobe JC-1 assay dye (Thermofisher, Waltham, MA, USA, # T3168) on PYR-treated cells using BD FACS Lyric Flow Cytometer. U87MG and LN-18 cells (~1 × 10^6^) were seeded in six-well culture plates and incubated at 37 °C and 5% CO_2_. Post 24 h incubation, U87MG cells were treated with PYR at 1, 2.5, and 5 µM and LN-18 at 0.5 and 1 µM for 16 and 24 h respectively. CCCP (carbonyl cyanide 3-chlorophenylhydrazone) at 50 µM served as a positive control. Following this treatment, the cells were stained with 2 µM JC-1 dye for 30 min. Post-JC-1 staining, the cells were washed with PBS and centrifuged at 1000 rpm for 5 min to remove excess dye. The cell pellet was re-suspended in 500 µL of PBS with a gentle flick. Further, 10,000 cells from each sample were acquired on a flow cytometer to quantify J-aggregates and J-monomers.

### 2.6. Reactive Oxygen Species (ROS) Assay

ROS generation in U87MG and LN-18 cells was analyzed using CellROX^®^ Deep Red flow cytometry assay kit (Molecular Probes, Eugene, OR, USA, # C10422). U87MG and LN-18 cells (~1 × 10^6^) were seeded in six-well culture plates and incubated at 37 °C and 5% CO_2_. Post 24 h incubation, cells were treated with PYR at different concentrations. After PYR treatment, the cell pellet was obtained by centrifugation at 400× *g* for 10 min. Finally, cells were resuspended in serum-free DMEM media. A ROS inducer TBHP (tert-butyl hydroperoxide) was used as a positive control. Cells were treated with 200 µM TBHP for 45 min. Further, cells were stained with 10 nM of CellROX Deep Red reagent and incubated for 45 min at 37 °C protected from light. Cells were washed twice using PBS. Stained cells were acquired on a flow cytometer for the CellROX Deep Red reagent (Ex-640/Em-660/10 nm BP).

### 2.7. Colony Formation Assay

U87MG and LN-18 cells were seeded in six-well plates (500–700 cells/well) and incubated for 24 h prior to compound addition. PYR and Doxorubicin alone or in combinations at different concentrations were added in wells and incubated for 48 h. After 48 h incubation, media containing compounds were replaced with fresh media, and plates were further incubated for 7–10 days. The colonies formed were fixed with acetic acid:methanol mixture (1:6) for 1 h followed by staining with 0.5% crystal violet. Pictures were captured on an Olympus microscope at a 10X eyepiece. Visible colonies from the respective wells were quantified using Image J software (V1.52a).

### 2.8. Wound Healing Assay

LN-18 cells were seeded in 24 well-plates (3 × 10^5^ cells/well) and further incubated for 24 h in complete media. Once confluent, scratches were drawn in each well with a sterile 200 µL tip across the center of the well. After scratches were generated, wells were washed twice with fresh medium to remove detached cells. Wells were replenished with fresh medium containing different concentrations of PYR (0.5 and 1 µM) and Doxorubicin (10 nM) alone or in combination. Distance between the edges of the wound was measured at 0 and 24 h post-drug addiction. Images were captured at 10X. Captured images were further quantified using Image J software to measure the closure of the wound.

### 2.9. PYR Combination Studies

For combination studies, U87MG and LN-18 cells (4000 cells per well) were seeded in 96-well plates and incubated at 37 °C and 5% CO_2_ for 24 h. Post incubation, both the drugs were added simultaneously, and plates were incubated for 72 h. The dose for PYR was selected based on its IC_50_ values in both the cell lines. PYR was taken at two or three suboptimal doses (≤IC_50_) and Doxorubicin was used at a dose range of 1 nM to 1 µM for the studies. The aim of the study was to observe the shifting of the sigmoidal curve towards the left axis with the increasing concentration of PYR, thereby decreasing the IC_50_.

PYR was also combined with Ritonavir to find out the cytotoxic effect of PYR at suboptimal concentrations (≤ IC_50_: 1, 3, and 5 µM) and Ritonavir (10 and 30 µM) alone or in combination was observed in U87MG cells treated for 72 h. The effect of drug combination was evaluated using the bliss independent analysis method to determine the index of synergy.

### 2.10. Western Blot

Protein expression was confirmed by Western blotting. RIPA buffer (Sigma #RO278) with proteases and phosphatases inhibitors was used to generate cell lysates after treatment with PYR. Total protein content in the lysates was quantified with the Pierce BCA protein assay kit (#23227). Proteins in cell lysates (around 5–50 μg) were separated on 7.5–12% SDS polyacrylamide gels by electrophoresis and transferred by dry method to a PVDF/nitrocellulose membrane. The membranes were blocked for an hour at RT with 5% non-fatty milk (Bio-Rad, Hercules, CA, USA, #1706404) solution in PBS containing 0.1% Tween (PBST). The membranes were incubated with various primary antibodies against different proteins. The blots were probed with primary antibodies incubated overnight, followed by 3 washes with PBST. The membranes were then incubated with appropriate secondary antibodies (1:10,000 or 1:20,000) for 1 h at RT. After 3 washes with PBST, the immunoblots were developed using enhanced luminol-based chemiluminescent substrate (ECL) substrate and the images were visualized using the ChemiDoc XRS system (Version 6.1, Bio-Rad, Hercules, CA, USA).

### 2.11. Par-4 Levels from Human Plasma

For analysis of Par-4 levels in clinical samples, study approval was taken by NCCS, Pune, from the Institutional Ethics Committee and informed consent was obtained from all the volunteers. Blood (5 mL) was processed for plasma separation and 2.5 µL plasma from normal healthy volunteers and GBM patients was checked for Par-4 levels by western blot and quantification using Image J software.

### 2.12. Par-4 Induction

U87MG cells were seeded in six-well plates (1 × 10^6^ cells/well) in DMEM complete media and treated with PYR (0.5 and 1 µM). The Par-4 induction at mRNA level was checked at 6, 18, and 24 h. Whereas, the Par-4 induction at the protein level was observed at 24 h. For Par-4 mRNA expression the cells were harvested, and RNA was extracted using RNA Xpress (HiMedia, India, #MB601) as per the manufacturer’s instructions. cDNA was synthesized and used as a template for RT-PCR. The PCR fragments were run on agarose electrophoresis, images captured using Bio-Rad Gel doc system, and quantified for densitometry analysis using Image J software. For Par-4 protein expression, the cells were harvested in 2X lamelli buffer (200 µL), heat-denatured at 100 °C, and snap chilled on ice. The Par-4 protein expression was evaluated with Par-4 primary antibody (Santa Cruz Biotech, Dallas, TX, USA, US # R-334) by Western Blotting.

### 2.13. Quantitative Real-Time PCR

GBM cells, U87MG, and LN-18 were seeded at 1.0 × 10^6^ cells/well in six well-plates and treated with PYR (0.5, 1, and 2.5 µM). Cells were harvested at different time points (2, 6, and 24 h) and RNA was extracted using TRIZOL (Invitrogen, Carlsbad, CA, USA). The first-strand cDNA was synthesized (2 μg of total RNA) using Superscript III reverse transcriptase enzyme (Invitrogen, Carlsbad, CA, USA) and oligo deoxythymidine (dT) as primers (Sigma Aldrich, Bangalore, India) (Appendix A). Quantitative real-time PCR was performed using SYBR Green Supermix (Thermofisher, Waltham, MA, USA) as per the manufacturer’s instructions. The relative mRNA expression of selected genes was calculated by the ΔΔ threshold cycle (Ct) method.

### 2.14. siRNA Mediated Silencing of Par-4

To elucidate the mechanism of action of PYR, U87MG was taken as a model cell line for siRNA experiments. U87MG cells (2.5–5 × 10^5^ cells/well) were plated in six-well plates in complete media. The cells were transfected with 30 pmol of Par-4 siRNA (Thermofisher, Waltham, MA, USA) using Lipofectamine RNAiMAX transfection Reagent (#4392420, Invitrogen, Waltham, MA, USA) as described by the manufacturer. The siRNA–Lipofectamine complex was added dropwise to the cells and incubated at 37 °C for 6 h. Post transfection, media was replaced with a complete DMEM medium, and the cells were harvested at 2, 4, 6, 18, and 24 h for mRNA expression and 24, 48, and 72 h for protein expression of Par-4, Cyclin D1 (#2922, Cell Signaling, Danvers, MA, USA) and Ki67 (#ab15580, Abcam, Cambridge, UK). GAPDH served as a loading control (#MA5-15738, Invitrogen, Waltham, MA, USA). Further, the siRNA-mediated Par-4 silenced cells were incubated with PYR (1 and 2.5 µM) for 6, 18, and 24 h for analyzing the differential expression of mRNA and proteins of Par-4, Cyclin D1, and Ki67.

### 2.15. Statistical Analysis

Statistical difference between the groups was determined by One-way analysis of variance (ANOVA) and post hoc multiple variances using a Tukey test (IBM SPSS Statistics_V20). For drug combination studies, the efficacy of different drugs alone/combinations is expressed as modeled T/C value, which is the actual measured T/C values on a scale ranging from 0 to 1, where 1 corresponds to a T/C value of 100%. The bliss neutral value is the product of modeled T/C of individual drug concentrations. The difference between the bliss neutral value and the modeled T/C value for the various combination was taken as the bliss index. The bliss index is arranged on a scale ranging from −1.0 to 1.0. Positive values (Bliss Index ≥ 0.15, blue) indicate synergy, negative values (Bliss Index ≤ −0.15, red) indicate antagonism, and zero is the neutral value (white).

## 3. Results

### 3.1. In Vitro Cytotoxicity of PYR in GBM Cells

In vitro cytotoxicity of PYR was determined in a panel of six glioma cell lines with diverse genetic backgrounds (Appendix A). These cell lines included TMZ sensitive and resistant cells, patient-derived xenograft (PDX) as well as glioma stem cells. PYR shows IC_50_ in the low micromolar range in all the cell lines tested. (Figure 1A–C). Of particular significance is its activity against LN-18 cells (IC_50_—1.16 µM), which express a high level of MGMT and are known to be TMZ-resistant. PYR was found to be equally effective in TMZ-sensitive cells viz. U87MG, LN-229, SBN-19, and U-251 with IC_50_ of 6.82, 2.3, 2.9, and 3.4 µM in, respectively (Figure 1A). In addition, PYR exhibited in vitro anti-tumor activity in patient-derived xenograft CNXF-2599L cells (IC_50_—3.6 µM) (Figure 1A) and LN-18 and U87MG spheroids with IC_50_ of 5.70 and 12.58 µM respectively (Figure 1B). As PDX cells retain most of the characteristics of parental patient tumors, so the efficacy response observed in PDX cells and 3D spheroids would have a high degree of probability to get replicated in clinical settings. Interestingly, PYR was also shown to be effective in glioma stem cells with IC_50_ of 1.8 and 2.9 µM in G1 and HNGC-2 cells respectively as discussed in Section 3.9. All cytotoxicity data generated for PYR on GBM cells in this study were compared with the available literature for PYR in different cancer types (Appendix A). Interestingly, PYR was positioned better in rank as an anti-proliferative agent as determined by in vitro cytotoxicity assays in GBM (Figure 1D,E) [14]. Figure 1C shows comparative IC_50_ data for PYR in TMZ resistant and TMZ sensitive cells.

### 3.2. PYR Induces FTIC-Annexin V Stained Early Apoptosis in GBM Cells

The likely mechanism by which PYR induces a cytotoxic effect on tumor cells was investigated using flow cytometry. The cell cycle distribution analysis was carried out using an asynchronous population of GBM cells. U87MG when treated with PYR at 1, 2.5, 5, and 7.5 µM represented a dose and time-dependent increase in the apoptotic population. Sub-G1 population at <IC_50_ (5 µM) was 25.45% at 72 h post PYR incubation (Figure 2A). LN-18 cells after treatment with PYR at 0.1, 0.5, 1, 1.5, 2, and 2.5 µM showed a dose and time-dependent increase in the sub-G1 population (Figure 2D). Profound increases in sub-G1 levels (81% and 90% at 48 and 72 h respectively) were observed at 2X IC_50_ of PYR. With the time, increases in the sub-G1 levels in both cell lines indicated a major portion of the cell undergoing apoptosis after PYR treatment. LN-18 cells appear to be more sensitive to PYR treatment, even though LN-18 cells grow in vitro as bipolar or stellate cells with pleomorphic nuclei with a doubling time of 72 h along with a high-level expression of Bcl-2 protecting these cells from the extrinsic apoptotic pathway.

Additionally, dose-dependent treatment of U87MG and LN-18 with PYR displayed Annexin V FITC-annexin V+/PI− population of cells, indicating the occurrence of early apoptosis (Figure 2B,E). In U87MG cells, early apoptotic events were 17 and 53% after PYR treatment at <IC_50_ (5 µM) at 24 and 48 h respectively. Treatment of LN-18 at <IC_50_ (1.5 µM) showed 21 and 55% early apoptotic events post 24 and 48 h incubation. Similarly, Annexin V FITC-annexin V+/PI+ population of cells, indicating late apoptotic was also seen to be increased with PYR dose (Appendix A). In LN-18 cells, late apoptosis events were found to be 10 and 20% at 1.5 µM PYR after 24 and 48 h treatment Importantly, increased apoptosis by PYR was accompanied by significant (*p* < 0.001) downregulation of mRNA levels of proliferation markers (Ki67, Cyclin D1, and DDX3) and hypoxia marker (HIF-1α) (Figure 2C,F; Raw data for Figure 2C–F can be found in Appendix A). In both the cell lines, the data obtained at the mRNA transcript level clearly indicated dose and time-dependent inhibition of Ki-67 and Cyclin D1 (Figure 2C,F). These findings are in line with our data that PYR induces cell cycle arrest and inhibition of proliferation as depicted by flow cytometry analysis (Figure 2D,E,G). Changes in cell cycle phases viz. G1, S1, and G2M were also noted in both GBM cell lines (Figure 2G and Appendix A). Figure 2H shows reduced protein expression of Ki-67 and Cyclin D1 in U87MG cells after 24 h of PYR treatment.

### 3.3. PYR Promotes Mitochondrial Depolarization (ΔΨM) and Generates ROS in GBM Cells

The effect of PYR on mitochondrial membrane depolarization was studied in U87MG and LN-18 cells using JC-1 dye. U87MG cells were treated with 1, 2.5, and 5 µM PYR for 16 h, while LN-18 cells were treated with 0.5 and 1 µM PYR for 24 h. PYR was found to cause mitochondrial depolarization in a dose-dependent manner as indicated by the formation of high levels of J-monomer. While the levels of J-monomer in LN-18 cells were 6.05 and 23.11% at 0.5 and 1 µM respectively, in U87MG the levels were 2.7, 14.2, 19.3% at 1, 2.5, and 5 µM respectively (Figure 3A,B).

The ability of PYR to induce ROS was studied in U87MG cells treated for 6 h and showed dose-dependent ROS positive cells 2.8, 4.4, 14.4, 53.5, 58.3, and 69.0% respectively (Figure 3C,D). Whereas 1.6, 15.6, 25.7, 36.1, 46.3 and 83.6% ROS positive population was observed in LN-18 cells treated with PYR at 0, 0.5, 1.0, 1.5, 2 and 3 µM for 4h respectively (Figure 3E,F). The ROS induction in both the cell lines was evident from the intensity shift of CellRox deep dye with an increase in PYR dose and a proportionate decrease in the live cell population (Figure 3D–F).

### 3.4. PYR Sensitizes GBM Cells to Doxorubicin Treatment

Combination studies were performed to check the ability of PYR to sensitize GBM cells to Doxorubicin treatment. The dose for PYR was selected based on its IC_50_ values in both the cell lines. When PYR was combined with Doxorubicin, enhanced cytotoxicity was observed in a dose-dependent manner in the GBM cell lines (Figure 4A–D). The sigmoidal curve for Doxorubicin alone drifted toward the left side when combined with PYR at 72 h (Figure 4A,C). The effect of this two-drug combination was evaluated using the bliss index (Appendix A). In U87MG cells, treatment of PYR at 2.5 and 1.5 µM with Doxorubicin displayed synergy (BI > 0.15) and slight synergy (BI between 0–0.15) respectively (Figure 4B). IC_50_ value shifted from 0.1 µM to 0.039 µM when PYR (2.5 µM) was used as a sensitizing agent for Doxorubicin treatment (Figure 4A). Similar observations were noted in LN-18 cells where the combination with PYR (1.5 µM) flattened the sigmoidal curve of Doxorubicin indicating synergy of the combination. The shift in IC_50_ value was noted from 1.16 to <0.05 µM (Figure 4D). In addition, PYR was assessed for its ability to sensitize glioma cells to Doxorubicin treatment using an anchorage-dependent colony-forming assay (CFA). U87MG and LN-18 cells treated with PYR alone showed a dose-dependent inhibition of colonies. Importantly, a combination of PYR (500 nM) with Doxorubicin (10 nM) in U87MG and PYR (0.1 µM) and Doxorubicin (10 nM) in LN-18 displayed significant inhibition of colony formation compared to drug alone (Figure 4E,F). Corroboratively, it is postulated that PYR restores sensitivity in MDR cells through strong hydrophobic interaction with P-gp that reduces drug efflux and increases intracellular concentration [18].

### 3.5. PYR Prevents Cell Migration in TMZ Resistant LN-18 Cells

The nature of GBM cells to invade the surrounding brain parenchyma is a major challenge to the effective use of radiotherapy and chemotherapy. Therefore, the potential effect of PYR on cancer cell migration was tested using an LN-18 cell line. PYR alone (500 and 1000 nM) and in combination with Doxorubicin (10 nM) was added in 24-wells where scratches were created on the confluent monolayer of LN-18 cells. After 24 h treatment, the results demonstrated that PYR alone could inhibit the migration of LN-18 cells in a dose-dependent manner (Figure 5A). Importantly, PYR also demonstrated a significant (*p* < 0.001) anti-migration effect when combined with Doxorubicin at 10 nM (Figure 5A,B).

Notably, epithelial to mesenchymal markers (Slug, SUMO2, β-catenin, and E-cadherin) were studied to establish an in-detailed mechanism of action of PYR in both GBM cells (Figure 5C,D). Treatment with PYR showed a statistically significant, *p* < 0.05) inhibitory effect at 24 h in a dose-dependent manner for the transcripts of β-catenin, Slug, and SUMO2. The significant upregulation of the transcript for E-cadherin was noted at 24 h, indicating the restoration of mesenchymal to epithelial (MET) trait after PYR treatment.

### 3.6. Ritonavir Synergizes PYR Treatment in U87MG Cells

Ritonavir is an approved protease inhibitor being used as an anti-retroviral agent in combination with other medications to treat HIV/AIDS. Besides a strong CYP3A4 inhibitor, it is being considered an adjuvant treatment agent for GBM [31]. Ritonavir at 10 and 30 µM sensitizes PYR treatment at 1, 3, and 5 µM in U87MG at 72 h (Figure 6A). Photomicrographs of Ritonavir treatment with PYR distinctly exhibited apoptotic population and changed morphology in U87MG cells (Figure 6B). Two drug combinations were evaluated using bliss independence analysis. The difference between the Bliss neutral value and the modeled T/C for the combination was taken as an index of synergy. The bliss index was 0.35 and 0.5 at PYR (5 µM) with 10 and 30 µM of Ritonavir respectively indicating synergy of combination (Figure 6C). It is also reported that co-administration of PYR with Ritonavir would substantially increase exposure to Ritonavir as PYR may inhibit CYP2D6 metabolism and Pgp efflux transport of Ritonavir [32].

### 3.7. PYR Induces Tumor Suppressor Par-4

To determine whether tumor suppressor Par-4 is induced by PYR, Par-4 expression was evaluated both at mRNA and protein levels. U87MG cells were treated with 1 and 2.5 µM PYR for 6, 18, and 24 h to check for mRNA transcript levels. The results obtained revealed that Par-4 mRNA transcript levels were increased on treatment with PYR (Figure 7A). Similar results were depicted for Par-4 protein, which was evaluated by western blot analysis. The U87MG cells treated with PYR (0.5 and 1 µM) for 24 h could effectively upregulate Par-4 protein levels (Figure 7B). The notable finding of PYR to induce Par-4 is a unique mechanism that could be effectively used as a potential approach to treat resistant GBM.

### 3.8. Silencing of Endogenous Par-4 in U87MG

Further, to evaluate the Par-4 mediated mechanism of PYR, U87MG as a model cell line was transiently transfected with either Par-4 siRNA (30 pmol) or negative control siRNA (NC siRNA). The Par-4 mRNA levels were evaluated 2, 4, 6, 18, and 24 h post silencing. The fold difference for Par-4 mRNA transcript revealed that effective silencing was noted from 6 h onwards till 24 h (Figure 7C). Negative control did not show reduced levels of Par-4 transcript at any time point. The silencing of Par-4 was also studied at protein level after 24, 48, and 72 h post-transfection. The Par-4 expression was analyzed by western blot and the densitometry results obtained are shown in Figure 7D. The fold difference for Par-4 expression was 0.23, 0.06, and 0.02 at 24, 48, and 72 h post-transfection respectively. Further to ascertain whether Par-4 induction is mediated by PYR, Par-4 silenced U87MG cells were treated with PYR (1 and 2.5 µM) for 24, 48, and 72 h post silencing to determine mRNA levels of Par-4 and proliferation markers viz. CyclinD1 and Ki67. It was observed that upon treatment of Par-4 silenced U87MG cells with PYR, proliferation makers viz. cyclin D1 and Ki67 mRNA levels were not restored up to 72 h as against Par-4 silenced control (Figure 7E). These findings suggest that the PYR-specific mechanism certainly facilitates a Par-4 mediated anti-cancer effect in GBM cells in vitro. It was noteworthy to observe that Par-4 levels were increased in Par-4 silenced U87MG cells treated with 1 and 2.5 µM PYR. Simultaneously at the protein level a marginal increase in Par-4 and decrease in Ki67, Cyclin D1 was noted (Figure 7F–H).

### 3.9. PYR Is Effective against Glioma Stem Cells

In vitro cytotoxicity of PYR was determined in highly tumorigenic glioma stem-like cells G1(primary) and HNGC-2 (Stem-like) cells. PYR exhibited IC_50_ of 2.9 µM and 1.8 µM in HNGC-2, and G1 respectively (Figure 8A). Further, PYR was tested for its ability to upregulate intracellular, secretory Par-4 and GRP-78 in human LN-18, HNGC-2, and G1. Cell lysates were assessed for expression of intracellular Par-4 and GRP-78, while secretory Par-4 was measured in supernatants of cells post 24 h PYR addition. The data obtained clearly indicates that PYR effectively upregulated intracellular and secretory Par-4 in HNGC-2 and G1 cells. The levels of cleaved Par-4 were significantly upregulated as against vehicle control in both the cell lines. Importantly, PYR could increase the expression of GRP-78 (Figure 8B). Similarly, the effect of PYR on stemness makers viz. Oct-4, Nanog, and Sox-2 were studied in surrogate GBM cells to delineate probable mechanistic effects in cancer stem cells. The heatmap depicts modulation of stem cell markers in both GBM cells post 6 and 24 h PYR treatment (Figure 8D). GAPDH served as a loading control.

### 3.10. Par-4 Levels in Normal and GBM Patients

Par-4 levels in a clinical sample from a normal healthy volunteer (*n* = 37) and GBM patients (*n* = 11) were determined using western blot analysis (Figure 9A–C). The Par-4 levels in plasma from GBM patients were significantly lower than the normal volunteers as evident from the scatter plot (Figure 9C). The data from the study has provided valuable information on the expression of Par-4 in normal volunteers and glioma samples. The current approach for the induction of a naturally occurring tumor suppressor (Par-4), which is down-regulated in GBM patients, could be a better therapeutic option.

## 4. Discussion

GBM is highly aggressive in nature and despite the advancement in standard therapies, the prognosis of GBM remains extremely poor. Thus, GBM treatment is an area of high unmet need. Due to the nature of the disease, GBM has a high clinical trial failure rate leading to many discontinued therapies. Most of the drugs that have shown promise in early-phase, single-armed trials unfortunately fail in larger, randomized studies. According to the global data, most of the current pipeline drugs in the phase III trials are either immunotherapies, peptide vaccines, checkpoint inhibitors, dendritic cell vaccines, or small molecule inhibitors. The two agents that are considered the most promising, are depatuxizumab mafodotin and DCVax-L. However, both therapies have challenges; depatuxizumab mafodotin has toxic ocular side effects, while DCVax-L, being a personalized therapy, is very challenging to deploy at a higher scale [19,33]. Given the high attrition rates, substantial costs, and slow pace of new drug discovery and development, repurposing ‘old’ drugs to treat both common and rare diseases is increasingly becoming an attractive proposition because it involves the use of de-risked compounds, with potentially lower overall development costs and shorter timelines.

In this study, we report for the first time the use of an antimalarial drug, PYR as a potential therapeutic agent against GBM. PYR is a benzonaphthyridine derivative, recently approved in combination with artesunate for the treatment of uncomplicated malaria and its safety profile is well established [12]. Interestingly, PYR has also shown anti-cancer potential in pre-clinical studies against several cancers [14]. Villanueva et al. demonstrated PYR-triggered cytotoxic activity by induction of apoptosis in different cancer cell lines [17].

Our study evidently indicates that PYR causes a significant anti-proliferative effect in GBM and patient-derived (PDX) cell lines with IC_50_ values in the low micromolar range. Other studies have reported the anticancer activity of PYR in various cancer cell lines with IC_50_ values in the range of 1.5–21.4 µM (Appendix A) [17,18]. Based on our findings, PYR is positioned better in rank against GBM as compared to other cancers (Figure 1D,E). Importantly, PYR inhibited the proliferation of both U87MG and LN-18 cells that are genetically distinct. U87MG cells are p53 wild type, PTEN mutated, expressing low levels of MGMT, while LN-18 cells are p53 mutated, PTEN wild type, expressing high levels of MGMT and MDR1 It is well known that high tumor MGMT and P-gp expression are associated with TMZ resistance [34,35]. Interestingly, our study demonstrates that LN-18 cells are most sensitive to PYR with an IC_50_ of 1.16 µM. Furthermore, PYR significantly downregulated the mRNA expression of proliferation and cell cycle markers Ki67, DDX3, and Cyclin D1.

PYR being a bona fide DNA-intercalating agent and an inhibitor of DNA topoisomerase II, induces DNA damage and cell death [14]. Our study demonstrates that PYR-induced apoptosis is associated with mitochondrial membrane depolarization and upregulation of cleaved caspase 3. Further, PYR could generate high levels of ROS in U87MG and LN-18 cells. Recent studies have reported that many chemotherapeutic drugs generate high levels of ROS in cancer cells and induce apoptosis via disruption of the mitochondrial membrane function [36]. Overproduction of cellular ROS causes damage to DNA, RNA, proteins, and cell cycle progression, which subsequently leads to apoptotic cell death [37].

GBM represents a high unmet need with a limited number of therapeutic agents available for treatment, therefore, it is imperative to look at potential drugs, which can be used in combination. Interestingly, doxorubicin-based nanoparticles have been reported as a novel approach to treating multidrug resistance (MDR) glioma cells [38,39,40] and improving survival in intracranial glioma orthotopic models [41]. Taken together, we considered the use of doxorubicin in combination with PYR as the best strategy to treat GBM cells in in vitro studies. Combination studies indicated that PYR sensitized LN-18 and U87MG cells to Doxorubicin as observed in the cytotoxicity, colony formation, and cell migration studies. This data is in line with the earlier report demonstrating the synergistic activity of PYR and Doxorubicin against multidrug-resistant (MDR) K562/A02 and MCF-7/ADR human cancer cells [42]. It is well known that ABC efflux transporters such as P-gp, MRP, and BCRP limit the central distribution of drugs that are beneficial to treating GBM as these efflux transporters contribute to the BBB function. Modulation of these transporters forms a novel strategy to enhance the penetration of drugs into the brain thus yielding better treatment options. Recent studies have reported that high expression of Cyclin D1 is associated with elevated P-gp levels and poor survival in GBM patients [43]. Significantly, GBM cells exposed to PYR displayed a reduction in Cyclin D1 expression and inhibition of cell proliferation. It is also known that inhibition of P-gp overcomes the effects of Cyclin D1 overexpression, in turn sensitizing GBM cells to TMZ-induced apoptosis [44]. It is therefore possible that the considerable decrease in the expression of Cyclin D1 in GBM cells is attributed to the function of PYR as a P-gp inhibitor [42]. It is noteworthy that Ritonavir an anti-retroviral drug and a P-gp inhibitor is in phase I clinical trial in combination with TMZ, a P-gp substrate for recurrent GBM (NCT02770378) [31]. In this regard, our data also indicates that the combination of PYR and Ritonavir exhibited profound synergistic anti-proliferative activity in U87MG cells. It would be interesting to further explore this combination in animal models to warrant its potential. Additionally, preliminary analysis conducted by us using a computational blood-brain barrier algorithm indicates that PYR has the potential to pass through the BBB (Appendix A).

Another important aspect is cancer stem cells, which are a small sub-population of cells within tumors with capabilities of self-renewal and differentiation. Similarly, GSC contributes to a small population of cells in GBM tumors playing a crucial role in recurrence and conferring resistance to drugs including TMZ [45]. It is noteworthy that our present research demonstrates the anti-proliferative activity of PYR against GSC viz, HNGC-2, and G1 with an IC_50_ of 2.9 µM and 1.8 µM respectively. Interestingly, PYR also reduced the stemness-associated markers viz, OCT-4, Nanog, Sox-2, and invasiveness of GBM cells. These changes were associated with the downregulation of EMT markers such as Slug, SUMO2, and β-catenin, concurrently upregulating the expression of E-Cadherin. These properties of PYR enhance its potential for the treatment of GBM and it will be interesting to understand the mechanistic pathway involved in the action of PYR against GSC.

One of the important features of any potential anti-cancer agent is to differentially modulate the expression of tumor suppressors and/or oncogenes. Par-4, a tumor suppressor protein, can selectively cause apoptosis in a wide variety of cancers without affecting normal cells [46]. However, Par-4 is known to be down-regulated in several cancers [46]. Therefore, agents that can induce Par-4 from either normal and/or cancer cells are an attractive therapeutic option. In this study, we unraveled a yet unidentified mechanism of PYR involving Par-4. PYR induces secretory and intracellular Par-4 from GBM cells. Intriguingly, PYR also enhanced the expression of GRP-78, a binding partner of Par-4. High expression of GRP-78 confers tumor-specific selectivity for Par-4 facilitated anticancer activity. The effectiveness of Par-4 inducing ability of PYR was confirmed in siRNA-mediated Par-4 knockdown cells. Remarkably, we observed that Par-4 silencing resulted in the up-regulation of proliferation markers Cyclin D1 and Ki67. Treatment of the silenced cells with PYR led to the down regulation of these markers and upregulation of Par-4. This is an exceptional finding and relates well with our observation that the plasma Par-4 levels in the GBM patients are significantly down-regulated as compared to healthy volunteers. Therefore, the use of PYR in GBM patients could lead to upregulation of Par-4 and cancer cell-specific activity in an autocrine as well as paracrine manner. Globally, research on Par-4 has advanced from the bench to the bedside with two clinical trials have been completed for solid tumors. Proposedly, Par-4 is emerging as a therapeutic target for clinical translation, with ample scope for population-based studies.

The multiple activities of PYR in cancer cells that modulates cellular pathway to induce growth inhibition and apoptosis of glioma cells impacting the crucial functions in GBM have been summarized in Figure 10. PYR upon intercalating with DNA in p53 mutated LN-18 cells causes upregulation of ROS, thereby leading to mitochondrial membrane depolarization, and initiates caspase 9 and 3 mediated intrinsic apoptosis pathways. Meanwhile, in p53 wild type U87MG cells, PYR along with upregulating ROS also activates p53 followed by cell cycle arrest mediated by the repression of Cyclin D1 through P-gp inhibition. PYR significantly induces the expression of both cellular and secretory Par-4 from normal and cancer cells. Nuclear translocation of intracellular Par-4 causes subsequent inhibition of NF-κB, which further leads to inhibition of proliferation, invasive properties, and EMT markers. It is also known that intrinsic Par-4 can cause permeabilization and depolarization of the mitochondrial membrane, leading to the release of apoptogenic factors such as cytochrome c, AIF, Smac/DIABLO, etc. Once released, these factors activate the caspase signaling cascade and orchestrate the cell death process [47]. Further, Par-4 secreted in response to PYR treatment binds to GRP-78 and initiates caspase-3 mediated cell apoptosis.

## 5. Conclusions

We strongly believe that emerging knowledge on Par-4 unlike other tumor suppressors has positioned Par-4 as an important target against multiple cancers including GBM. Our study suggests that repurposing PYR, used in conjunction with standard of care may serve as an effective therapeutic option in GBM (Figure 11). It seems clear from the data that manipulating the Par-4 pathway by PYR will undoubtedly bring considerable therapeutic benefits for GBM. Par-4 is known to be dysregulated in several cancers including GBM, and patient data supports this observation. However, this needs to be further validated using a larger patient pool. It is reasonable to suggest that Par-4 levels can become a valuable tool as an anti-cancer agent as well as an important biomarker for designing patient-specific therapy not only in GBM but also for other cancers making Par-4 a potential theranostic agent.

## Figures and Tables

**Figure 1 cancers-14-03198-f001:**
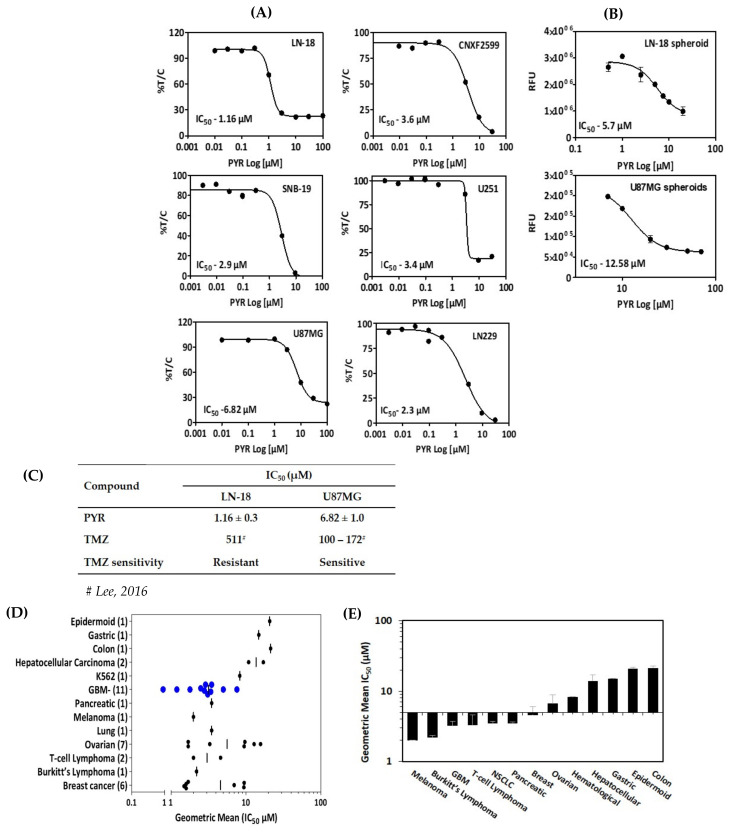
(**A**) In vitro cytotoxicity (IC_50_) of PYR in a panel of six GBM cell lines at 72 h. CNXF-2599 is a patient-derived xenograft (PDX) cell line isolated from a male patient with unknown differentiation. Cytotoxicity is expressed as % survival (% T/C) as a ratio of treated by control. (**B**) Activity of PYR in 3D multicellular spheroids (MCS) or gliospheres generated from GBM cell lines. LN-18 and U87MG cells were seeded in triplicates in 96 well-plates (4000 cells/well) with 1.5% agarose layered at the bottom. Nascent gliospheres were further incubated for 24 h at 37 °C, 5% CO_2_. On day 2, gliospheres were treated with PYR (0.1–100 µM) for 6 days. At the end of incubation, Alamar blue (10 µL) was added in wells, incubated in dark for 3–4 h, and fluorescence (RFU) was read at 530/590 nm. (**C**) Table shows a comparison of IC_50_ values for PYR and TMZ in TMZ resistant (LN-18) and TMZ sensitive (U87MG) cell lines. (**D**) Geometric mean of IC_50_ values for GBM cell lines (N = 11) compared with literature-based IC_50_ values for 12 other cancer types. Numbers in parenthesis represent the number of samples used to determine the geometric mean. Blue circles represent the geometric mean of 11 GBM cell lines. The vertical black line represents the geometric mean for a given set of cell types. (**E**) Histogram represents in vitro activity of PYR in various cell types arranged with low to high sensitivity rank order of geometric means (IC_50_).

**Figure 2 cancers-14-03198-f002:**
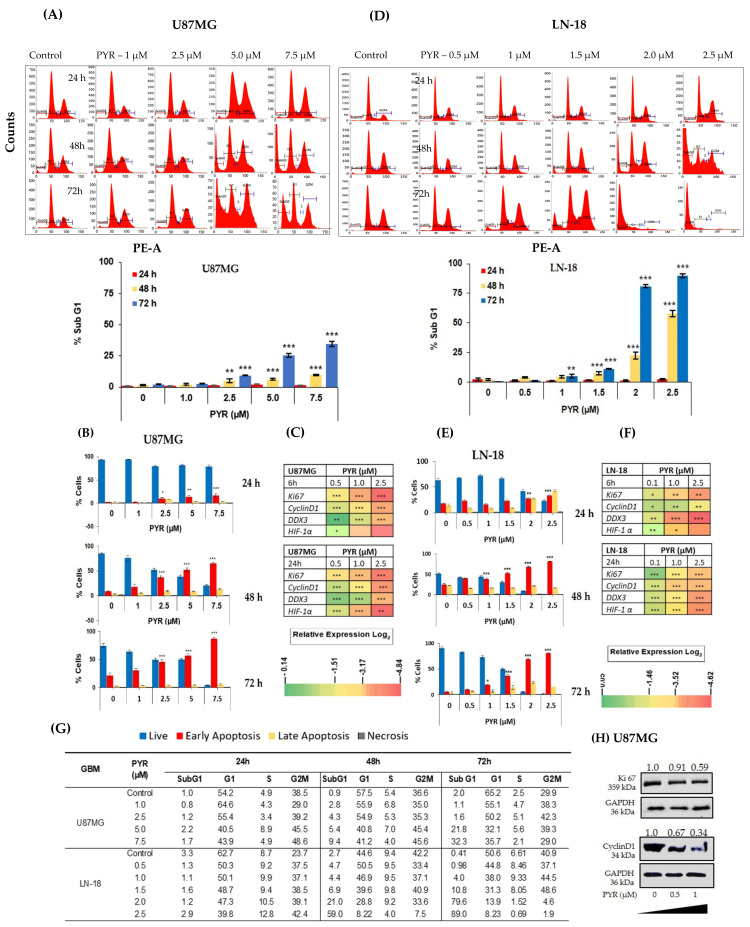
(**A**,**D**) PYR induced dose and time-dependent apoptosis (% sub-G1) in U87MG and LN-18 at 24, 48, and 72 h. Ten thousand cells were acquired and further analyzed by BD FACS-Lyric flow cytometry. Histograms represent increased sub-G1 levels in both GBM cells after treatment with PYR. (**B**,**E**) Histograms represent annexin-V early apoptosis in GBM cells in a time and dose-dependent manner after PYR treatment. Cells were treated with PYR at various concentrations at 24, 48, and 72 h. GBM cells were double-stained with Annexin V-FITC and PI and analyzed using flow cytometry. (**C**,**F**) Heatmap represents modulation of proliferation (Cyclin D1, Ki67, and DDX3) and hypoxia (HIF-1α) markers in U87MG and LN-18. The relative expression of genes was calculated with the relative ΔΔCt method, using GAPDH as a housekeeping gene for normalization. Data shown here are mean ± SD (*n* = 2). Statistical differences between the groups were determined by one-way ANOVA and post hoc multiple variance test using Tukey. The statistically significant difference is represented as * *p* ≤ 0.05, ** *p* ≤ 0.01 and *** *p* ≤ 0.001 for control v/s specific group. (**G**) The table shows DNA histograms with the percentage of cells in each phase of the cell cycle after treatment with PYR from a representative experiment. (**H**) Western blot and densitometry analysis of Ki67 and Cyclin D1 expression in U87MG treated with PYR at indicated concentrations for 24 h. GAPDH served as a loading control. The whole Western blot can be found in Appendix A.

**Figure 3 cancers-14-03198-f003:**
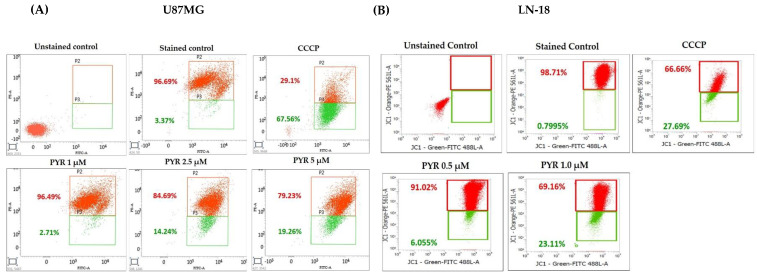
(**A**,**B**): Representative graphs for PYR-induced mitochondrial depolarization using JC-1 dye. U87MG and LN-18 (1 × 10^6^ cells/mL) were treated with PYR at indicated concentrations for 16 and 24 h respectively. After treatment, cells were stained with JC-1 dye and fluorescence output was measured at 490/590 nm. CCCP (carbonyl cyanide 3-chlorophenylhydrazone) was used as a positive control. Cells were subjected to flow cytometric analysis by measuring the depolarized population of GBM cells (green color population) post PYR treatment in the lower quadrant using BD FACS Lyric. (**C**–**F**) Dose-dependent ROS induction in U87MG and LN-18 cells post 6 and 4 h PYR treatment respectively. The ROS levels were measured by determining Cell ROX Deep Red fluorescence intensity via flow cytometry at an emission/excitation of 640/665 nm. (**D**–**F**) The histogram depicts the trend of Live cells (red bar) and ROS-positive cells (blue bar) in GBM cells post PYR treatment. The data shown are mean ± SD (*n* = 2). Statistical analysis was performed using one-way ANOVA followed by a post hoc by Tukey test. The statistically significant difference for control v/s specific group is represented as * *p* ≤ 0.05, ** *p* ≤ 0.01 and *** *p* ≤ 0.001.

**Figure 4 cancers-14-03198-f004:**
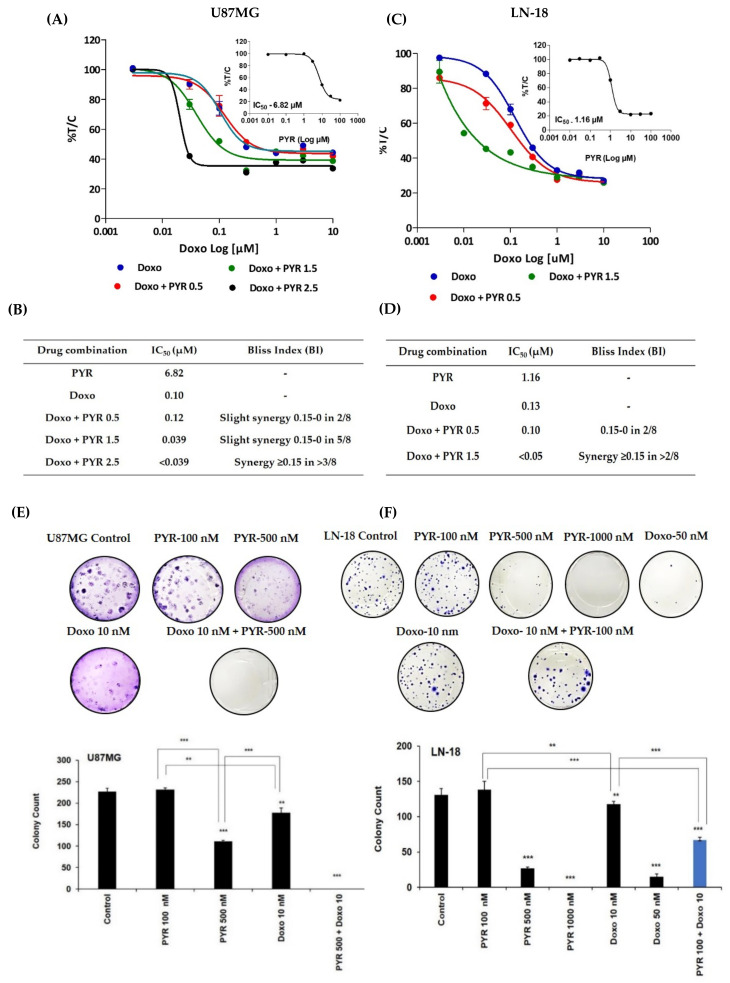
(**A**,**C**) Sigmoidal curve for combination treatment in U87MG and LN-18 cells at indicated concentrations for 72 h. Treatment with PYR at different concentrations with Doxorubicin shifts the sigmoidal curve to the left side indicating synergism. (**B**,**D**) Bliss index is shown as the difference between the expected T/C value (Bliss neutral) and the measured T/C value (modeled T/C) on a scale ranging from −1.0 to 1.0. Positive values (Bliss Index ≥ 0.15, blue) indicate synergy, negative values (Bliss Index ≤ −0.15, red) indicate antagonism, and zero is the neutral value (white). (**E**,**F**) PYR alone or in combination with Doxorubicin inhibits anchorage-dependent colony formation in U87MG and LN-18 cells. Histograms represent the quantification of stained colonies using Image J software. The data shown are mean ± SD, (*n* = 2). A statistically significant difference between the control v/s different doses or within the groups of PYR was determined by one-way ANOVA and post hoc multiple variance by a Tukey test. The statistically significant difference is represented as ** *p* ≤ 0.01, and *** *p* ≤ 0.001 for control v/s specific group or within the groups.

**Figure 5 cancers-14-03198-f005:**
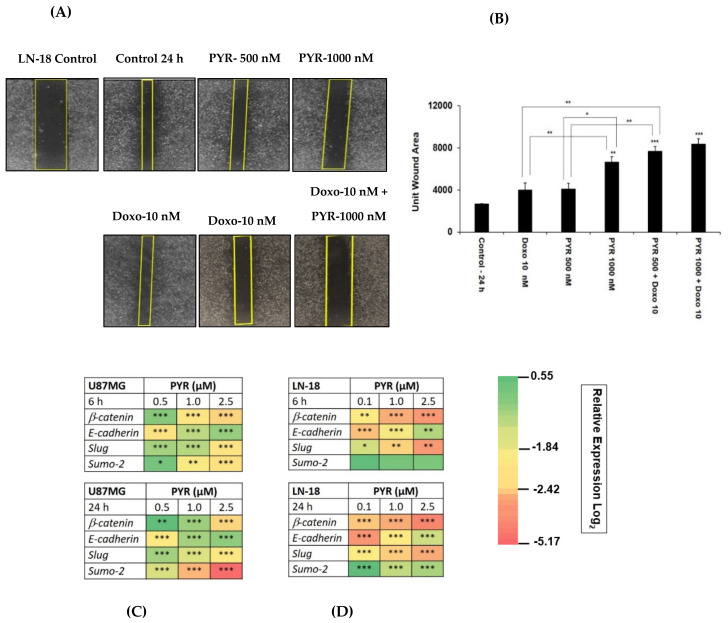
(**A**) A wound healing assay representing the effect of PYR (500 nM and 1000 nM) alone and in combination with Doxo (10 nM) on migration and invasive potential of LN-18 cells 24 h post addition of compounds. (**B**) Bar graphs show wound closure quantified by Image J software. (**C**,**D**) Heatmap depicts modulation of EMT markers in LN-18 and U87MG post 6 and 24 h treatment with PYR. The relative expression of genes was calculated with the relative ∆Δct method, using GAPDH as a housekeeping gene for normalization. The data shown represent the mean ± SD, (*n* = 2). Statistical differences between the groups were determined by one-way ANOVA and post hoc multiple variance using a Tukey test. The statistically significant difference is represented as * *p* ≤ 0.05, ** *p* ≤ 0.01, and *** *p* ≤ 0.001 for control v/s specific group or within the group.

**Figure 6 cancers-14-03198-f006:**
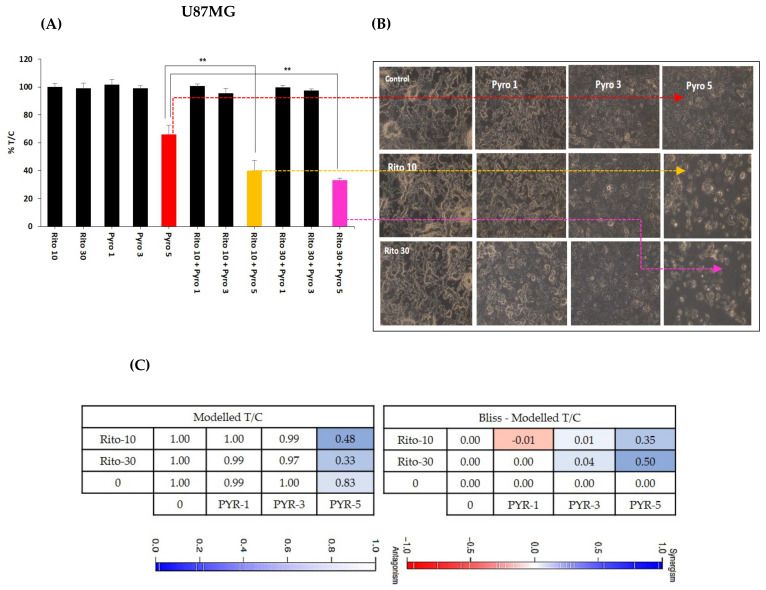
(**A**) Histograms showing the combined effect of Ritonavir (10 and 30 µM) with PYR at 1, 3, and 5 µM in U87MG, ***p* ≤ 0.01. (**B**) Photomicrographs showing apoptotic population and changed morphology in U87MG cells when treated with PYR and Ritonavir. (**C**) A combination of PYR and Ritonavir was evaluated using BI analysis. Bliss index is shown as the difference between the expected T/C value (Bliss neutral) and the measured T/C value (modeled T/C) on a scale ranging from −1.0 to 1.0. Positive values (Bliss Index ≥ 0.15, blue) indicate synergy, negative values (Bliss Index ≤ −0.15, red) indicate antagonism, and zero is the neutral value (white).

**Figure 7 cancers-14-03198-f007:**
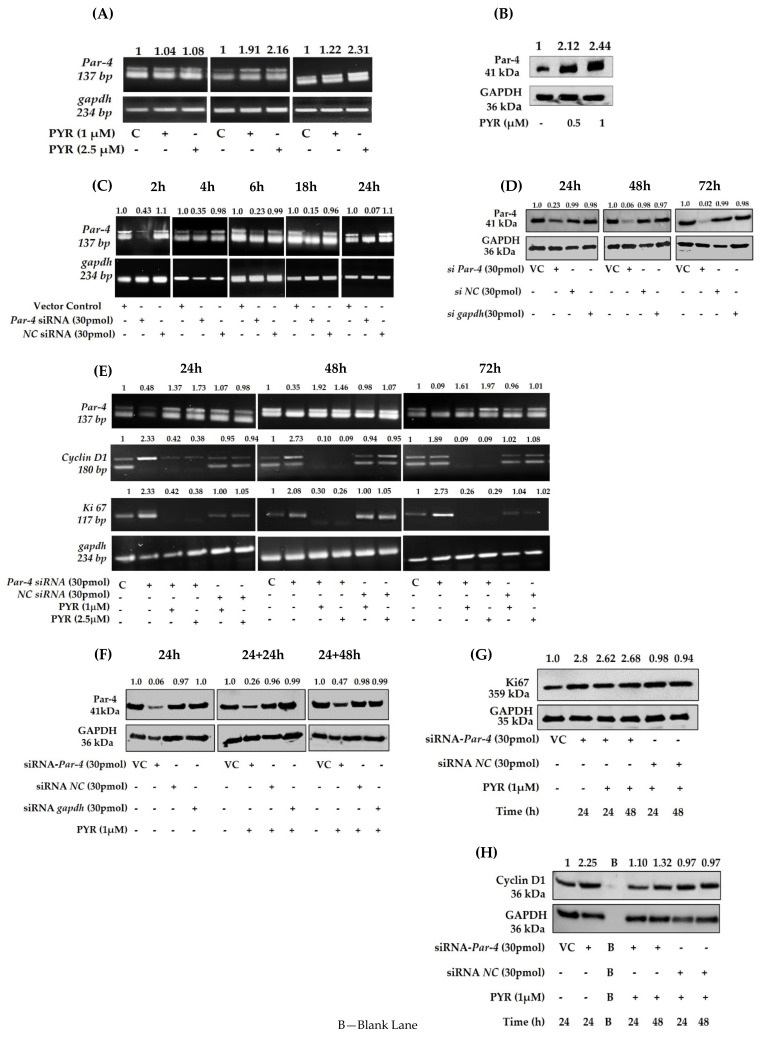
**(A**) PYR (1 and 2.5 µM) induced Par-4 mRNA transcript levels in U87MG cells 6, 18, and 24 h post-treatment. GAPDH was used as a loading control. (**B**) Par-4 protein expression levels in U87MG cells after PYR treatment at 0.5 and 1 µM post 24 h. Figures on top indicate fold difference for Par-4 expression determined after densitometry analysis using Image J software. (**C**) Silencing of Par-4 mediated by Par-4 specific siRNA (30 pmol) in U87MG at different time intervals viz. 2, 4, 6, 18, and 24 h. Scrambled siRNA served as a negative control. GAPDH served as a loading control. (**D**) Level of Par-4 protein expression up to 72 h in U87MG cells after silencing. (**E**) Increased mRNA levels of proliferation markers viz. Ki67 and Cyclin D1 after Par-4 silencing at 24, 48, and 72 h in U87MG cells. Reduced mRNA transcript levels after addition of PYR (1 and 2.5 µM) to silenced U87MG cells. (**F**) Par-4 protein levels 24 and 48 h after the addition of PYR (1 µM) to siRNA (30 pmol) silenced U87MG cells for 24 h. Scrambled SiRNA served as a negative control. GAPDH served as a loading control. (**G**,**H**) Reduced levels of proliferation markers viz. Ki67 and Cyclin D1 at 24 and 48 h after addition of PYR (1 µM) to silenced U87MG cells.

**Figure 8 cancers-14-03198-f008:**
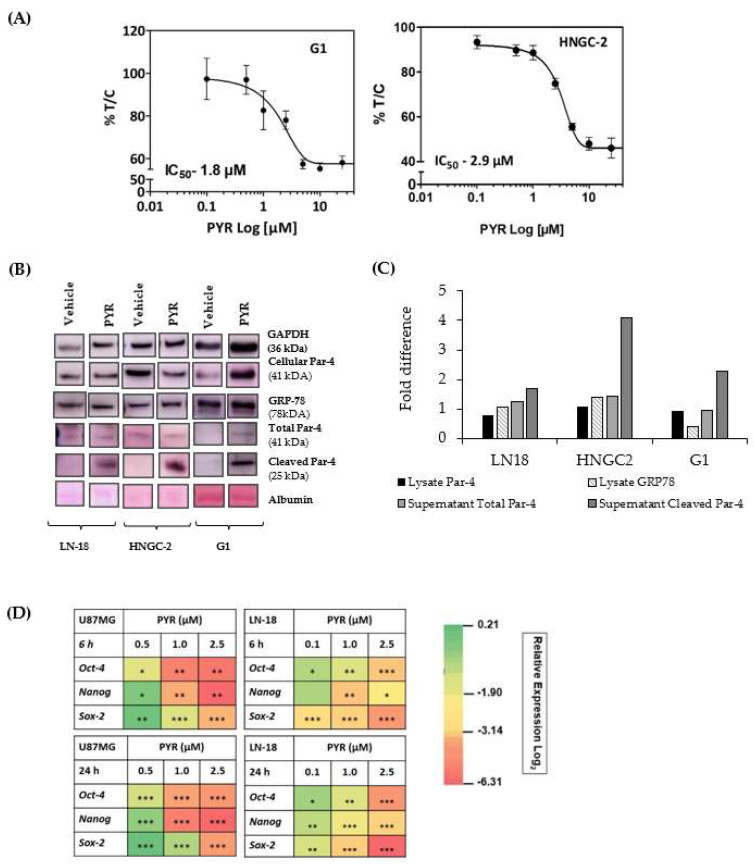
(**A**) Anti-proliferative activity of PYR in highly tumorigenic glioma stem-like G1 and HNCG-2 cells at 72 h. (**B**,**C**) PYR (25 µM) was tested for its ability to upregulate intracellular Par-4, secretory Par-4, and GRP-78 in three GBM cell lines. WB was done with cell lysates for assessing the expression of intracellular Par-4 and GRP-78 while secretory Par-4 was measured in supernatants of cells treated with the test compounds for 24 h. Untreated cells and DMSO-treated cells served as controls and vehicle controls respectively. The intensity of bands was measured by performing densitometric analysis using Image J software. The data was interpreted as a fold change with respect to vehicle control. GAPDH was used as a loading control. (**D**) Heatmap depicts modulation of stem cell markers in U87MG and LN-18 post 6 and 24 h treatment with PYR. The relative expression of genes was calculated with the relative ΔΔCt method, using GAPDH as housekeeping gene for normalization. A statistically significant difference between a control v/s and a PYR treated group was determined by one-way ANOVA and Post hoc multiple variance by a Tukey test. * *p* ≤ 0.05, ** *p* ≤ 0.01 and *** *p* ≤ 0.001.

**Figure 9 cancers-14-03198-f009:**
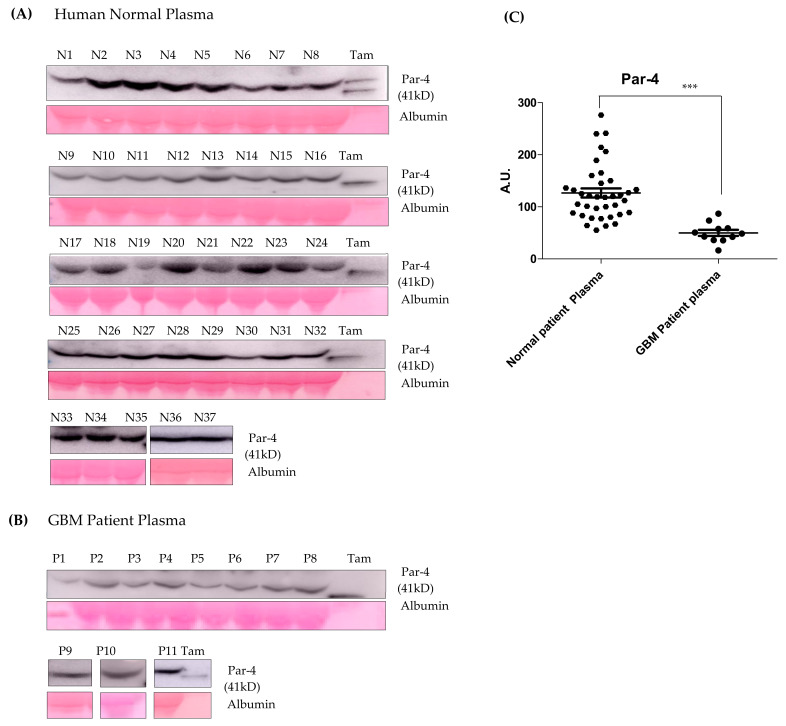
(**A**,**B**) The western blot for Par-4 estimation in the plasma samples of normal volunteers and GBM patients using an Image J software. Albumin and Ponceau stained blots were considered for normalization. (**C**) Scatter plot representing average Par-4 levels in normal and GBM patient plasma. Tamoxifen (Tam) treated cell- supernatant was used as a positive control. Statistical significance for Par-4 levels between normal and GBM patients was determined using ANOVA followed by the post hoc Tuckey test (*** *p* ≤ 0.001).

**Figure 10 cancers-14-03198-f010:**
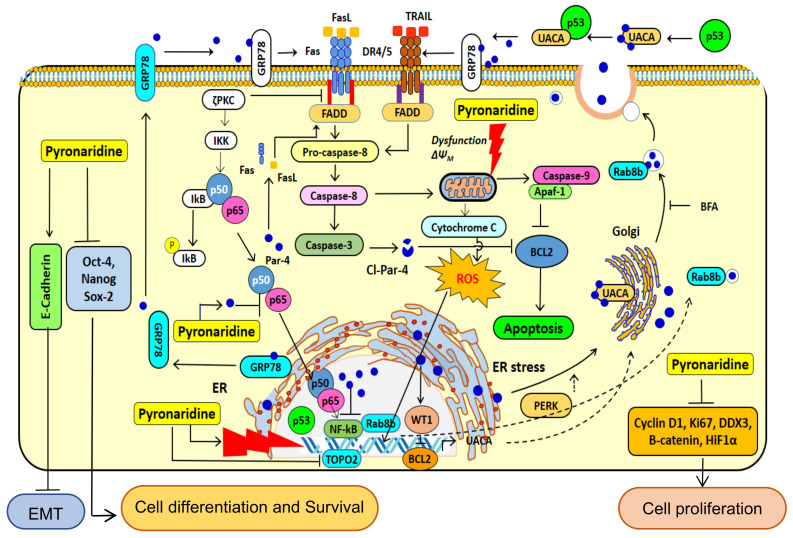
Proposed cellular mechanism of action of PYR. The illustration depicts the proposed mechanism of PYR in cancer cells, modulating cellular pathways to induce growth inhibition and apoptosis in glioma cells. PYR is an acridine derivative, which intercalates DNA and impairs DNA repair pathways. PYR-induced DNA damage promotes activation of p53-mediated cell cycle arrest. In addition, PYR inhibits Topoisomerase II and induces mitochondrial depolarization thereby initiating caspase 3 and 9 mediated intrinsic apoptosis pathways. PYR induces tumor suppressor protein—Par-4 through inhibition of nuclear localization of p65, NF-κB, BCl-2 and activating pro-apoptotic pathway proteins BAX. PYR increases oxidative stress after DNA damage and decreases MMP. PYR upregulates the production of p53, capase3/9, Bax, and Cyto-C and downregulates the production of Bcl-2, promoting the release of Cyto-C, increasing caspase-3/7 activities, and potentiating apoptosis in glioma cells. Bax, Bcl-2-associated X protein; Bcl-2—B-cell lymphoma 2; Cyto C—cytochrome c.

**Figure 11 cancers-14-03198-f011:**
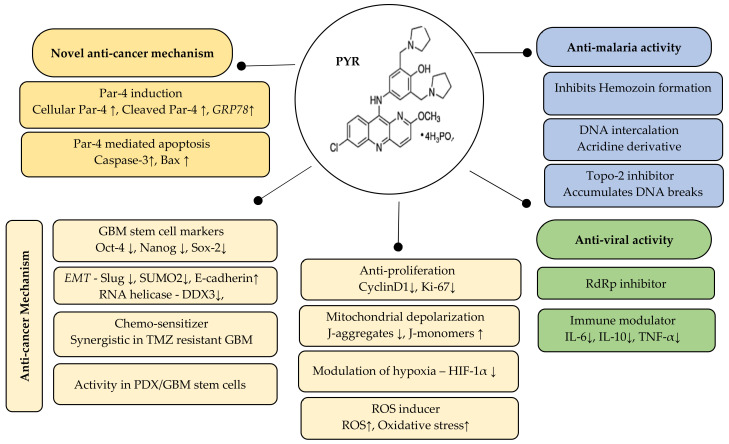
Summary of the therapeutic potential of PYR: Par-4 induction and Par-4 mediated apoptosis is a novel mechanism of PYR. The anti-cancer mechanism includes inhibition of stemness, EMT and hypoxia markers besides mitochondrial depolarization and induction of ROS. The right panel indicates the known anti-malarial and anti-viral activity of PYR. ↓ indicates downregulation and *↑* indicates upregulation.

## Data Availability

The datasets analyzed during the current study are available from the corresponding author upon request.

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
