# Peer review of "Prostate Apoptosis Response-4 (Par-4): A Novel Target in Pyronaridine-Induced Apoptosis in Glioblastoma (GBM) Cells"

_cancers, 2022, doi:10.3390/cancers14133198_

Round 1

Reviewer 1 Report

The paper has been properly improved by the authors and I haven't any scientific concern to declare. The methods are clear and the experimental plan well established. The results give an adjunct value to the field of anti-GBM therapy so I think that the contribution might be suitable for final publication

Reviewer 2 Report

.

This manuscript is a resubmission of an earlier submission. The following is a list of the peer review reports and author responses from that submission.

Round 1

Reviewer 1 Report

The manuscript entitled as “Prostate apoptosis response-4 (Par-4): a novel target in Pyronaridine-induced apoptosis in glioblastoma (GBM) cells” is well designed and written. However, I have few suggestions which may help to improve the quality of the manuscript are as follows:

  1. I suggest evaluating the expression of Ki-67 and Cyclin D1 in LN-18 cells after 24 hours of

PYR treatment.

  1. I recommend analyzing the expression patterns of the cell cycle checkpoints after PYR treatment at 2.5, 5 and 7 uM concentration against U87MG cells.
  2. Discuss the rationale behind the PYR and doxorubicin combination treatment.
  3. How authors have decided the dosage of PYR and doxorubicin for the combination treatment?
  4. Densitometric analysis of the protein bands should be incorporated into the figures.
  5. Effect of Par-4 siRNA against PYR treatment should also be evaluated in LN-18 cells at different time intervals.

Reviewer 2 Report

The authors submitted a nice paper describing the anti-GBM effect of PYR.

The huge amount of data presented are convincing and statistically significant. The rationale is quite clear; indeed concerning this I have only a minor question. Why the authors focused their attention of Par-4 tumour suppressor? and not other ones? please clarify and better discuss it.

Furthermore Have the authors tried to corroborate the detection of plasma level of Par-4 through ELISA assay? in order to adjunct translational value to the clinical data, ELISA is a more suitable quantitative technique.

Reviewer 3 Report

General comments to the Authors

Despite the positive reviews of the original versions of the manuscript, there are glaring weaknesses that significantly diminish enthusiasm for its potential clinical utility. First, the study lacks requisite statistical power and replication to reliably validate the accuracy and reproducibility of its results and conclusions. Second, the study lacks requisite and validating control groups utilizing established Par-4 inhibitors to establish relative selectivity of action. Third, the study is largely confirmatory of a previously published study by Chem Biol Drug Des. 2022 Jan;99(1):83-91.; PLoS One. 2018 Nov 5;13(11):e0206467.; Clin Cancer Drugs. 2021 Mar;8(1):50-56. and therefore lacks significant novelty.

Round 2

Reviewer 1 Report

The revised file of the manuscript is greatly improved. However, due to the technical flaws I in the manuscript I think it should not be accepted in the present condition.

Authors have mentioned that the data presented in the manuscript demonstrates the mean ± SEM of at least three independent experiments with triplicates. Student’s unpaired t-test was employed when means of two groups were compared. Statistical significance was evaluated by calculating p values. Differences where p < 0.05 were considered statistically significant (*p < 0.05; **p < 0.01 and ***p < 0.001).
However, in most of the representative bar graphs, errors bars were hardly visible. Such as in the graph represented in figure 2, figure 3 B and D, 5B, 8B.

Similarly, the pcr data is represented as a heat map, which does not give enough information about the variation on the data value obtained from three triplicates of three independent experiments!

I doubt, how it is possible to get so tight mean ± SEM values in triplicates of three independent experiments?

If it was not reproducible at least three times in three independent experiments, it could be just an observation!

I strongly suggest to submit the evidence of three replicates of all the experimental datas of three independent experiments in the supplementary file along with the manuscript.